# Grounding Bodily Awareness in Visual Representations for Efficient Policy Learning

## Abstract

Learning effective visual representations for robotic manipulation remains a fundamental challenge due to the complex body dynamics involved in action execution. In this paper, we study how visual representations that carry body-relevant cues can enable efficient policy learning for downstream robotic manipulation tasks. We present **I**nter-token **Con**trast (**ICon**), a contrastive learning method applied to the token-level representations of Vision Transformers (ViTs). ICon enforces a separation in the feature space between agent-specific and environment-specific tokens, resulting in agent-centric visual representations that embed body-specific inductive biases. This framework can be seamlessly integrated into end-to-end policy learning by incorporating the contrastive loss as an auxiliary objective. Our experiments show that ICon not only improves policy performance across various manipulation tasks but also facilitates policy transfer across different robots. The project website: `https://anonymous.4open.science/w/ICon/`

## 1 Introduction

Vision serves not only the awareness of the external environment but also the awareness of one's own self (Gibson, 2002). Through vision, we perceive our bodies, monitor our movements, and maintain a perceptual boundary between self and non-self. This form of bodily awareness, commonly referred to as *visual proprioception* (Bermúdez, 2011), enables agents to respond to their own bodily dynamics in a flexible and adaptive manner. Such responsiveness is essential for planning and executing actions in tasks that require high-level action sensitivity, such as locomotion and manipulation (Gibson, 1966). Going further, incorporating such inductive biases, particularly those arising from the agent's body within the visual field, can be highly beneficial to learning policies for robotic tasks (Gmelin et al., 2023; Hu et al., 2021; Soter et al., 2018). With awareness of the position and movement of its own body, a robotic agent can efficiently learn structured agent-environment representations from raw pixel observations (Gmelin et al., 2023).

However, despite existing efforts in visuomotor policy learning, extracting body-aware information from high-dimensional images remains challenging, especially in end-to-end learning frameworks where visual encoders are jointly optimized with policy networks (Levine et al., 2016). Since both components share the same optimization objective, models can easily converge to bottlenecks that inadvertently filter out task-irrelevant cues, including visual signals related to the agent's body. This issue becomes even more pronounced when training data is deficient. To address this, one approach is to augment the policy loss with an agent-centric auxiliary objective (Gmelin et al., 2023; Pore et al., 2024). These methods typically involve reconstructing RGB observations or agent masks from latent representations to implicitly disentangle a robotic agent from its environment. While this strategy has proven effective across various tasks, we argue that the reconstruction loss can undermine the training stability of policy learning. This raises a key question: is there a more natural way to derive disentangled agent-environment representations from pixels without sacrificing model performance and training stability?

To this end, we propose **I**nter-token **Con**trast (**ICon**), a contrastive learning approach designed to extract agent-centric representations from the Vision Transformer (ViT) (Dosovitskiy et al., 2020), a high-capacity visual encoder widely utilized in robotic manipulation (Fu et al., 2024; Karamcheti et al., 2023; Radosavovic et al., 2023; Xiao et al., 2022). ICon applies contrastive learning to the ViT's token-level features, where features corresponding to the agent are pulled together, and are

contrasted against those corresponding to the environment, and vice versa. By explicitly decoupling agent-specific and agent-agnostic features, we implicitly encourage the model to focus on agent-relevant information, rather than information of the entire scene. We further introduce the following technical contributions to enhance the performance of ICon:

- We bring Farthest Point Sampling (FPS) (Qi et al., 2017) into 2D domains to sample keys from tokens for contrastive learning. By encouraging a wide spatial distribution of keys, FPS ensures that the selected features capture diverse and informative aspects of either the agent or the environment, maintaining a good representation of the overall structure.
- We propose a multi-level design that fuses inter-token contrastive losses from multiple layers of the ViT encoder, enabling a more complete disentanglement between the agent and its environment within the learned visual representations.

Through extensive experiments, we demonstrate that integrating ICon with Diffusion Policy (Chi et al., 2023), a state-of-the-art imitation learning algorithm, leads to consistent performance improvements across 8 manipulation tasks spanning 3 different robots from 2 benchmarks. Code, data, and videos can be found: `https://anonymous.4open.science/w/ICon/`

## 2 RELATED WORK AND BACKGROUND

### 2.1 VISUOMOTOR POLICY LEARNING

Training control policies that map visual sensory inputs directly to motor actions has been widely studied in reinforcement learning (RL) (Kostrikov et al., 2020; Levine et al., 2016; Yarats et al., 2021a) and imitation learning (IL) (Chi et al., 2023; Lee et al., 2024; Mandlekar et al., 2021; Shafiullah et al., 2022). Among all, several works have explored learning improved representations for visual control through auxiliary tasks. Dasari and Gupta (2021) leverage learned representations to predict the gripper's future location as a 2D keypoint in the image for debugging purposes, although they do not explicitly use this auxiliary objective for representation learning. Extending this line of work, Yarats et al. (2021b) couple a policy network with an autoencoder to reconstruct raw image pixels from the learned latent space, which has proven effective to improve the sample efficiency of RL algorithms. Building upon this idea, Gmelin et al. (2023) incorporate an additional autoencoder to reconstruct binary agent masks, yielding an agent–centric representation that facilitates policy transfer across different robots. More recently, Li et al. (2024) introduce the reconstruction approach to the reverse diffusion process (Ho et al., 2020), where a decoder reconstructs both pixel and state information from the intermediate representations of a U-Net model (Ronneberger et al., 2015) to enhance the performance of a diffusion-based policy (Chi et al., 2023). Our approach is similar to Laskin et al. (2020) and Zhu et al. (2022), which augment the policy objective with an auxiliary contrastive loss. However, instead of focusing on extracting task-relevant semantics from high-dimensional images, we aim to explicitly encourage the policy to learn agent-centric visual representations.

### 2.2 CONTRASTIVE LEARNING

Contrastive learning is a self-supervised learning paradigm to learn useful representations from high-dimensional data, such as natural language (Radford et al., 2021), images (Caron et al., 2021; Chen et al., 2020; He et al., 2020; Radford et al., 2021), and videos (Nair et al., 2022; Sermanet et al., 2018; Xu et al., 2023). It can be interpreted as training an encoder for a dictionary look-up task, whose goal is to pull the query closer to a positive key while pushing it away from all other negative keys. This is typically achieved by minimizing a contrastive loss (Chopra et al., 2005), which serves as an unsupervised objective function for training the encoder networks. Commonly used contrastive losses include Triplet loss (Schroff et al., 2015), N-pair loss (Sohn, 2016), Noise Contrastive Estimation (NCE) loss (Gutmann and Hyvärinen, 2010), and InfoNCE loss (Oord et al., 2018). In this paper, we adopt a variant of the InfoNCE loss proposed by Wang et al. (2022):

$$\mathcal{L}_{\text{InfoNCE}}(q, \mathcal{K}^+, \mathcal{K}^-) = \frac{1}{|\mathcal{K}^+|} \sum_{k^+ \in \mathcal{K}^+} -\log \frac{\exp\left(q \cdot k^+/\tau\right)}{\exp\left(q \cdot k^+/\tau\right) + \sum\limits_{k^- \in \mathcal{K}^-} \exp\left(q \cdot k^-/\tau\right)}, \quad (1)$$

where $q$, $\mathcal{K}^+$, and $\mathcal{K}^-$ denote the query, the set of positive keys, and the set of negative keys, respectively; $(\cdot)$ denotes the dot product; and $\tau$ is a temperature hyperparameter.

## 3 VISUALLY GROUNDED AGENT-CENTRIC REPRESENTATIONS

In principle, it is possible to integrate ICon with any visuomotor policy that uses vision transformers as visual encoders. In this section, we begin with an overview of the vanilla vision transformer, followed by a detailed explanation of the key design choices of ICon as well as its integration with a visuomotor policy network. An overview of ICon is shown in Figure 1.

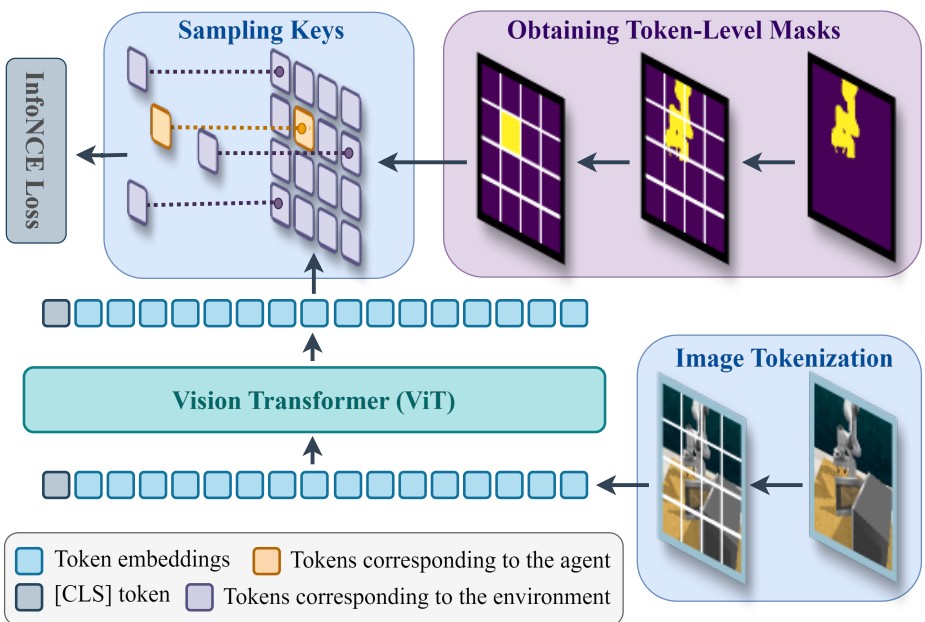

Figure 1: Overview of ICon. A full-scene RGB image containing a robotic agent is tokenized and processed by a vision transformer. The resulting token-level features (excluding the [CLS] token) are reshaped and aligned with a token-level mask derived from the agent's segmentation mask. Tokens corresponding to the agent and the environment are then sampled and used as keys to compute the inter-token contrastive loss.

### 3.1 PRELIMINARIES: VISION TRANSFORMERS

Vision Transformers (ViTs) (Dosovitskiy et al., 2020) extract token-level representations from high-dimensional images. As depicted in Figure 1, an image $\mathcal{I} \in \mathbb{R}^{H \times W \times 3}$ is first divided into non-overlapping patches, each of size $P \times P$, and then embedded into a sequence of tokens $\mathcal{T} \in \mathbb{R}^{N \times D}$, where $N = HW/P^2$ denotes the number of patches and $D$ is the embedding dimension. The token embeddings, prepended with a learnable classification token [CLS], are subsequently fed into the ViT encoder to produce a sequence of token-level features $[\mathcal{F}_{\text{cls}}, \mathcal{F}]$, where $\mathcal{F}_{\text{cls}} \in \mathbb{R}^D$ and $\mathcal{F} \in \mathbb{R}^{N \times D}$ correspond to the [CLS] token and the patch embeddings, respectively.

### 3.2 TOKEN-LEVEL AGENT MASKS

While we have obtained token-level features from the vision transformer, how can we determine which features are agent-specific and which are agent-agnostic? Recall that each token corresponds to an image patch consisting of a set of pixels. Each pixel can be classified as belonging to either the agent or the environment based on an agent mask (Gmelin et al., 2023; Hu et al., 2021; Pore et al., 2024). Therefore, we can propagate these pixel-level assignments to the token level.

Specifically, given the image $\mathcal{I}$ of the full scene, we use a segmentation model to generate a binary mask $\mathcal{M} \in \mathbb{R}^{H \times W}$, where $\mathcal{M}_{i,j} = 1$ for pixels occupied by the agent and 0 otherwise. This mask $\mathcal{M}$ is then patchified into $\mathcal{P}_{\text{mask}} = \{p_{k,l}\}_{k=1,l=1}^{H/P,W/P}$ following the same patchification procedure applied to the image $\mathcal{I}$ in ViT encoding. Since each patch $p_{k,l}$ may contain a mix of agent-related and

environment-related pixels, we determine its dominant class based on a masking threshold $\beta \in [0, 1]$: if the proportion of agent pixels in a patch exceeds $\beta$, the patch is considered agent-dominated and assigned a value of 1; otherwise, it is considered environment-dominated and assigned a value of 0 (Equation (2)). This yields a new patch-level (or token-level) mask $\mathcal{M}_{\text{token}} = \{m_{k,l}\}_{k=1,l=1}^{H/P,W/P}$, where $m_{k,l} \in \{0, 1\}$.

$$m_{k,\,l} = \begin{cases} 1 & \text{if } \text{sum}(p_{k,\,l}) > \beta P^2 \\ 0 & \text{otherwise} \end{cases}. \tag{2}$$

### 3.3 INTER-TOKEN CONTRASTIVE LOSS

Now that we have acquired the token-level features and the agent masks, we introduce an inter-token contrastive loss to help the model distinguish between the agent and its environment.

Our intuition is straightforward: features that belong to the same class (agent or environment) should be similar, while features coming from different classes should be dissimilar. To fulfill this, we encourage features of the same class to cluster together while enforcing separation between features of different classes, resulting in a clearer boundary between the agent and its environment in the learned feature space.

Specifically, given the token-level features $\mathcal{F}$ and the corresponding agent masks $\mathcal{M}_{\text{token}}$, we first rearrange the sequence-like features $\mathcal{F}$ into a 2D feature map $\mathcal{F}_{\text{map}} = \{f_{k,l}\}_{k=1,l=1}^{H/P,W/P}$ for subsequent processing. We then compute the agent-specific query $q_a$ and environment-specific query $q_e$ by averaging the corresponding features, as defined in Equation (3), where $\mathbb{I}(\cdot)$ stands for the indicator function. As for key selection, we adapt the Farthest Point Sampling (FPS) method (Qi et al., 2017) from point cloud sampling to the 2D domain (see Algorithm 1). Compared with random sampling, FPS promotes diversity through selecting points that are spatially well-distributed (see Figure 2), ensuring that the sampled keys capture diverse and representative features of the agent and the environment. By applying FPS within the feature map $\mathcal{F}_{\text{map}}$ while restricting the

---

**Algorithm 1** 2D Farthest Point Sampling

1: **Input:** 2D indices $\mathcal{V} = \{(k,l)\}_{k=1,l=1}^{H,W}$ , a binary mask $\mathcal{M} = \{m_{k,l} \in \{0,1\}\}_{k=1,l=1}^{H,W}$ indicating sampling regions, number of samples $N$ ($N \leq \sum m_{k,l}$)
2: **Output:** Indices of samples $\mathcal{V}'$
3: $\mathcal{D} \leftarrow \{d_{k,l} = \infty\}_{k=1,l=1}^{H,W}$ ▷ Distance map
4: Randomly select $(\tilde{k}, \tilde{l})$ where $m_{\tilde{k},\tilde{l}} = 1$
5: $\mathcal{V}' \leftarrow \{(\tilde{k}, \tilde{l})\}$
6: **for** $s = 1$ **to** $N - 1$ **do**
7:     $(\hat{k}, \hat{l}) \leftarrow \mathcal{V}'[-1]$
8:     **for** $k = 1$ **to** $H$, $l = 1$ **to** $W$ **do**
9:         $\hat{d}_{k,l} \leftarrow |\hat{k} - k| + |\hat{l} - l|$
10:         **if** $\hat{d}_{k,l} < d_{k,l}$ **then**
11:             Update $d_{k,l} \leftarrow \hat{d}_{k,l}$
12:         **end if**
13:     **end for**
14:     $(k^*, l^*) \leftarrow \arg\max_{k,l}(m_{k,l} \cdot d_{k,l})$
15:     $\mathcal{V}' \leftarrow \mathcal{V}' \cup \{(k^*, l^*)\}$
16: **end for**
17: **return** $\mathcal{V}'$

---

sampling regions using $\mathcal{M}_{\text{token}}$ and $(1 - \mathcal{M}_{\text{token}})$, we obtain a set of agent-specific keys $\mathcal{K}_a$ and a set of environment-specific keys $\mathcal{K}_e$, respectively. Note that for the agent-specific query $q_a$, the agent-specific keys $\mathcal{K}_a$ serve as positive keys, while the environment-specific keys $\mathcal{K}_e$ serve as negative keys, and vice versa for the environment-specific query $q_e$. Finally, we compute two symmetric InfoNCE losses (Equation (1)) for the queries using their respective positive and negative keys, and combine them together to form the ICon objective (Equation (4)). The complete pseudocode for ICon is provided in Algorithm 2.

$$q_a = \frac{1}{\text{sum}(\mathcal{M}_{\text{token}})} \sum_{k=1}^{H/P} \sum_{l=1}^{W/P} \mathbb{I}(m_{k,l} = 1) f_{k,l},$$

$$q_e = \frac{1}{\text{sum}(1 - \mathcal{M}_{\text{token}})} \sum_{k=1}^{H/P} \sum_{l=1}^{W/P} \mathbb{I}(m_{k,l} = 0) f_{k,l}. \tag{3}$$

$$\mathcal{L}_{\text{ICon}} = \mathcal{L}_{\text{InfoNCE}}(q_a, \mathcal{K}_a, \mathcal{K}_e) + \mathcal{L}_{\text{InfoNCE}}(q_e, \mathcal{K}_e, \mathcal{K}_a). \tag{4}$$

(a) Random sampling      (b) Farthest Point Sampling

Figure 2: Visualization of point distributions sampled from the agent mask. (a) Random sampling may result in points clustered within a small region. (b) Farthest Point Sampling (FPS) produces points that are well-distributed across the entire agent.

## 3.4 MULTI-LEVEL CONTRAST (MLC)

In the standard ICon formulation, inter-token contrastive learning is applied only at the final layer of the vision transformer. However, we argue that this is insufficient to fully decouple the agent and its environment within the visual representations. To achieve a more complete agent-environment disentanglement, we extend ICon to each transformer encoder layer (Vaswani et al., 2017) of the vision transformer. Specifically, let $\mathcal{F}^{(i)}$ represent the token-level output features (excluding the [CLS] token) from the $i$-th layer. The inter-token contrastive loss for this layer, $\mathcal{L}_{\text{ICon}}^{(i)}$, is computed as described in Section 3.3. The overall contrastive objective is then obtained by taking a weighted sum of the layer-wise contrastive losses:

$$\mathcal{L}_{\text{ICon}} = \sum_i \frac{\exp\left(\gamma \cdot i\right)}{\sum_i \exp\left(\gamma \cdot i\right)} \mathcal{L}_{\text{ICon}}^{(i)}. \qquad (5)$$

Here, $\gamma$ is a hyperparameter that controls the disentangling degree across transformer encoder layers. Prior work has shown that the shallow layers of a vision transformer primarily capture positional information, while deeper layers shift toward encoding more semantic features (Amir et al., 2021). This implies that shallower layers tend to produce more entangled agent-environment representations, resulting in larger inter-token contrastive losses. To strike a balance, we set $\gamma > 0$ to assign greater weights to the contrastive losses from deeper layers.

## 3.5 TRAINING

As described above, ICon enhances a policy's visual representations by introducing an agent-centric contrastive loss as an auxiliary objective during policy optimization. We utilize the widely adopted Diffusion Policy (Chi et al., 2023) to demonstrate how ICon can be incorporated into its training pipeline. Let $\mathcal{D} = \{(o_t \in \mathcal{O}, a_t \in \mathcal{A})\}_{t=1}^{T}$ denote a dataset consisting of observation-action pairs, where the observation space $\mathcal{O}$ comprises both image observations $\mathcal{I}$ and low-dimensional state information $\mathcal{S}$. Diffusion Policy learns a mapping $\pi : \mathcal{O} \to \mathcal{A}$ by training a visual encoder $\mathcal{E}$ jointly with a diffusion model (Ho et al., 2020) using a prediction loss $\mathcal{L}_{\text{pred}}$. In our framework, the visual encoder is instantiated as a vision transformer, whose output features $\mathcal{F}_{\text{cls}}$ and $\mathcal{F}$ are used to condition on the denoising diffusion process and compute the contrastive objective $\mathcal{L}_{\text{ICon}}$, respectively. By combining the prediction loss and the contrastive loss together with a weighting coefficient $\lambda$, we derive the following training objective for policy update:

$$\mathcal{L} = \mathcal{L}_{\text{pred}} + \lambda \mathcal{L}_{\text{ICon}}. \qquad (6)$$

In practice, we precompute the agent masks $\mathcal{M}$ and store them alongside the observations $o_t$ and actions $a_t$ in the dataset $\mathcal{D}$. During training, for each mini-batch sampled from $\mathcal{D}$, we apply identical image augmentations to the image observations and their corresponding masks before computing the training objective.

# 4 EXPERIMENTS

We conduct a systematic evaluation of ICon across **8** manipulation tasks spanning **3** robots from 2 simulation benchmarks. Through our experiments, we seek to answer the following questions:

**1)** To what extent does ICon improve the performance of the base policy?

**2)** What are the advantages of ICon over its counterparts?

**3)** Does ICon facilitate policy transfer across different robots?

**4)** What design choices of ICon have the most influence on its performance?

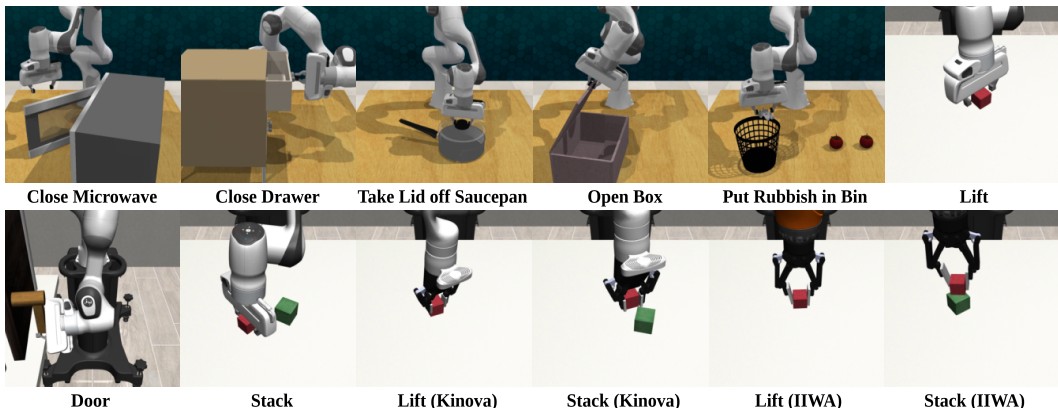

Figure 3: Visualization of simulated environments used for evaluation.

## 4.1 SIMULATION BENCHMARKS

*RLBench* (James et al., 2020): is a large-scale manipulation benchmark designed for meta learning, reinforcement learning, and imitation learning. It provides more than 100 robotic manipulation tasks ranging from simple target-reaching to complex long-horizon tasks. We select **5** tabletop tasks—*Close Microwave*, *Close Drawer*, *Take Lid off Saucepan*, *Open Box*, and *Put Rubbish in Bin*—which encompass object picking, articulated object manipulation, and long-horizon pick-and-place.

*Robosuite* (Zhu et al., 2020): is a widely used manipulation benchmark comprising 19 task environments that span both single-arm and dual-arm manipulation. From this benchmark, We select **3** representative tasks—*Lift*, *Door*, and *Stack*—which involve lifting a cube, opening a door, and stacking one cube on top of another, respectively.

## 4.2 DATASETS

We release a new dataset covering the 8 manipulation tasks across 3 different robots in the RLBench and Robosuite environments. In RLBench, data are collected using the built-in motion planning toolkit, whereas in Robosuite, data are collected via teleoperation. Specifically, we collect **50** human demonstrations per task using a Franka Emika Panda robot, and an additional **5** demonstrations each from a Kinova Gen3 robot and a KUKA LBR IIWA robot for the *Lift* and *Stack* tasks. Each human demonstration comprises a sequence of paired observations and actions, where observations include RGB images from two viewpoints (a third-person and a wrist-mounted camera) and robot proprioception (e.g., joint position, gripper status), and actions correspond to the end-effector poses. For each RGB image, we use the Segment Anything Model (SAM) (Kirillov et al., 2023; Ravi et al., 2024) to extract a segmentation mask of the robot in the scene, and store the robot mask alongside the observation-action pairs in the dataset, forming a sequence of observation-mask-action triplets. In the following experiments, we train different policies using the Franka-specific data for performance comparison and fine-tune the pre-trained policies on Kinova-specific and IIWA-specific data to evaluate few-shot policy transfer across robots.

## 4.3 EVALUATION SETUP

**Baselines.** We integrate and compare ICon with two variants of the Diffusion Policy (Chi et al., 2023): (i) **Diff-C**, a CNN-based variant that performs well on most manipulation tasks with minimal need for hyperparameter tuning; and (ii) **Diff-T**, a transformer-based variant shown to be particularly effective for complex manipulation tasks involving frequent action changes. We refer to our methods as **ICon-Diff-C** and **ICon-Diff-T**, respectively. Additionally, we compare against Crossway Diffusion (Li et al., 2024), which shares the same backbone as Diff-C but incorporates an auxiliary reconstruction loss to improve representation learning. For brevity, we refer to it as **Crossway-Diff-C**.

**Policy rollout.** Before each rollout, the simulated environment is randomly initialized using a predefined seed that is consistent across all learning algorithms. At each step, instead of relying solely on the current observation to predict the next action, the policy receives the past $T_o$ observations from the environment and predicts the next $T_a$ actions, of which only the first $T_a'$ are executed in the scene. In practice, we find it crucial to apply *Temporal Ensemble* (Zhao et al., 2023) to the predicted action sequences to ensure smoother control and mitigate action jitters.

**Evaluation methodology.** We report success rates for each learning algorithm and manipulation task. Results are averaged over 3 training seeds and 50 different environment initial conditions (150 episodes in total), with standard deviations computed across the 3 training seeds. A task is considered successful if and only if the reward returned by the simulated environment changes from 0 to 1. In addition, each task has a predefined maximum number of rollout steps; if the robotic agent fails to complete the task within this limit, the episode is deemed a failure.

Table 1: Performance comparison of different algorithms on the RLBench benchmark. We present success rates for 5 algorithms across 5 tasks in the format of (mean) $\pm$ (standard deviation), as described in Section 4.3.

|  | Diff-C | Diff-T | Crossway-Diff-C | ICon-Diff-C | ICon-Diff-T |
|---|---|---|---|---|---|
| Close Microwave | $0.040 \pm 0.016$ | $0.993 \pm 0.009$ | $0.033 \pm 0.019$ | $0.153 \pm 0.034$ | **1.000** |
| Close Drawer | $0.713 \pm 0.034$ | $0.893 \pm 0.025$ | $0.667 \pm 0.041$ | $0.713 \pm 0.050$ | **$0.913 \pm 0.047$** |
| Take Lid off Saucepan | $0.033 \pm 0.019$ | $0.280 \pm 0.075$ | $0.073 \pm 0.025$ | $0.113 \pm 0.050$ | **$0.413 \pm 0.151$** |
| Open Box | $0.087 \pm 0.074$ | $0.113 \pm 0.090$ | $0.047 \pm 0.066$ | **$0.300 \pm 0.043$** | $0.127 \pm 0.019$ |
| Put Rubbish in Bin | $0.000$ | $0.033 \pm 0.025$ | $0.000$ | $0.000$ | **$0.093 \pm 0.082$** |

Table 2: Performance comparison of different algorithms on the Robosuite benchmark. Success rates are reported for 3 algorithms across 3 tasks in the same format as in Table 1.

|  | Diff-C | Crossway-Diff-C | ICon-Diff-C |
|---|---|---|---|
| Lift | $0.527 \pm 0.104$ | $0.573 \pm 0.100$ | **$0.627 \pm 0.129$** |
| Door | $0.860 \pm 0.028$ | $0.827 \pm 0.082$ | **$0.887 \pm 0.034$** |
| Stack | $0.160 \pm 0.016$ | $0.067 \pm 0.025$ | **$0.220 \pm 0.016$** |

## 4.4 PERFORMANCE IMPROVEMENTS

As shown in Table 1, diffusion policies coupled with ICon consistently outperform or match the baselines across all 5 tasks in the RLBench simulated environments. Notably, ICon-Diff-C achieves absolute improvements of 21.3% and 11.3% over Diff-C in the *Open Box* and *Close Microwave* tasks, respectively. In another articulated object manipulation task *Close Drawer*, the positive effects of incorporating ICon are less pronounced, but ICon-augmented policies still perform on par with or better than the baselines. In contrast, Crossway-Diff-C underperforms Diff-C and ICon-Diff-C across all three articulated object manipulation tasks. In the *Take Lid off Saucepan* task, ICon-Diff-C and Crossway-Diff-C both exhibit higher success rates than Diff-C, with ICon-Diff-C showing more substantial improvements. Likewise, ICon-Diff-T surpasses Diff-T with an absolute improvement of

13.3%. In the long-horizon *Put Rubbish in Bin* task, all CNN-based diffusion policies fail to succeed, whereas ICon-Diff-T remains better than Diff-T.

As displayed in Table 2, ICon-Diff-C outperforms both Diff-C and Crossway-Diff-C across all tasks. In the *Open Door* task, Diff-C underperforms ICon-Diff-C but outperforms Crossway-Diff-C, aligning with earlier experimental results on articulated object manipulation tasks in the RLBench environments. In the *Stack* task, ICon-Diff-C surpasses both Diff-C and Crossway-Diff-C with improvements of 6.0% and 15.3%, respectively. Overall, integrating ICon into diffusion policies leads to improved performance across all 8 manipulation tasks.

## 4.5 TRAINING STABILITY

A key strength of ICon is to maintain good training stability during end-to-end policy learning. For a quantitative measure, we train each policy for an equal number of epochs with checkpoints saved every 50 epochs, and report the average of the top-10 success rates as well as the overall maximum success rate for the *Open Door* task. Results are visualized in Figure 4, with dark and light colors representing maximum and average success rates, respectively. The accompanying percentages stand for the relative drop from the maximum to the average performance. We see that when maximum performances are comparable, Crossway-Diff-C exhibits the largest gap between maximum and average success rates, indicating that the auxiliary reconstruction loss hinders the training stability of the base policy. In contrast, ICon-Diff-C shows superior training

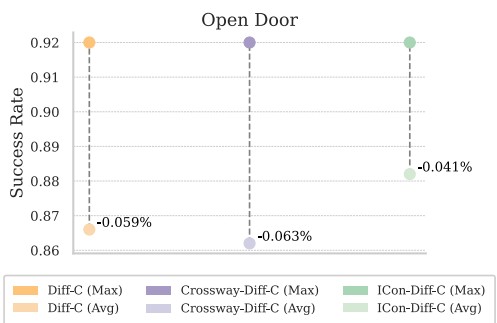

Figure 4: Comparison of training stability based on maximum and average performance during the training process.

stability by maintaining a relatively higher average performance throughout the training process. This suggests that ICon enables the base policy to learn more robust and consistent behaviors from pixel observations.

Table 3: Results of few-shot policy transfer across different robots on the Robosuite benchmark. Policies are transferred from a source robot to a target robot, with task success rates reported for each robot and learning algorithm. Success rates are displayed following the same format as in Table 1 and Table 2.

| Task | Source Robot | | Target Robot | | | |
|---|---|---|---|---|---|---|
| | Franka (Default Gripper) | | Kinova (Robotiq85) | | IIWA (Robotiq140) | |
| | Diff-C | ICon-Diff-C | Diff-C | ICon-Diff-C | Diff-C | ICon-Diff-C |
| Lift | $0.527 \pm 0.104$ | $\mathbf{0.627 \pm 0.129}$ | $0.233 \pm 0.066$ | $\mathbf{0.260 \pm 0.102}$ | $0.060 \pm 0.016$ | $\mathbf{0.100 \pm 0.125}$ |
| Stack | $0.160 \pm 0.016$ | $\mathbf{0.220 \pm 0.016}$ | $0.007 \pm 0.009$ | $\mathbf{0.053 \pm 0.025}$ | $0.007 \pm 0.009$ | $\mathbf{0.047 \pm 0.025}$ |

## 4.6 TRANSFERABILITY ACROSS ROBOTS

Here, we evaluate the transferability of ICon-augmented policies across 3 robots from the Robosuite benchmark, where variations come from both robotic arms (Franka, Kinova, IIWA) and grippers (Franka Default Gripper, Robotiq85, Robotiq140). We initially pre-train policies on data collected from a source robot, and then fine-tune them using a smaller dataset collected from a target robot. Results in Table 3 show that ICon enhances the performance of the base policy across all three robots in both the *Lift* and *Stack* tasks. We also find that polices are more effectively transferred to the Kinova robot than to the IIWA robot, which we believe is because of the appearance similarity between the Kinova and the source Franka robot.

## 4.7 ABLATION STUDY

We evaluate how each key component of ICon contributes to its overall performance. Specifically, we conduct ablation studies on: (i) the masking threshold $\beta$; (ii) the number of agent keys $N_a$ and environment keys $N_e$ used in computing the contrastive loss; and (iii) the key sampling and loss fusion strategies. These experiments are performed on the *Open Door*, *Close Microwave*, and *Open Box* tasks, respectively. A summary of the results is presented in Figure 5.

We note that choosing either $\beta < 0.5$ or $\beta > 0.5$ substantially impairs model performance, indicating that assigning equal weights (0.5) to agent-specific and environment-specific pixels during agent mask propagation yields the most accurate approximation of token-level masks. Next, we find that increasing the number of sampled keys beyond a certain point markedly extends training time. While a larger number of keys enables more effective disentanglement, a practical trade-off is achieved by setting $N_a = 10$ and $N_e = 50$. Finally, we observe that omitting Multi-Level Contrast (MLC) results in a noticeable decline in performance, which we attribute to the insufficient disentangling of intermediate representations in the vision transformer. A more significant performance degradation occurs when random sampling is applied in place of FPS for key sampling, likely due to the reduced expressivity of the sampled keys.

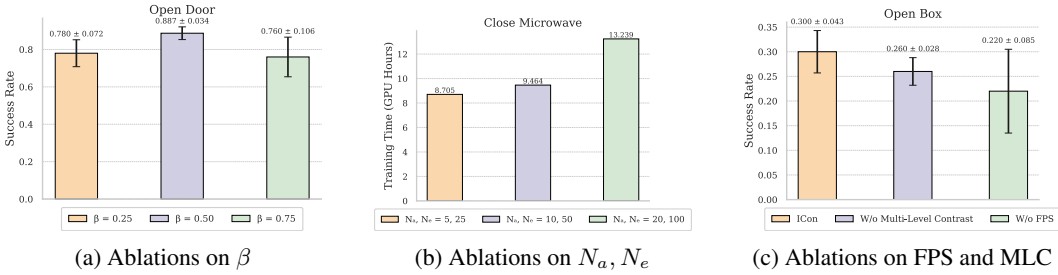

| (a) Ablations on $\beta$ | (b) Ablations on $N_a, N_e$ | (c) Ablations on FPS and MLC |

Figure 5: Summary of ablation experiments on (a) the masking threshold $\beta$, (b) the number of agent keys $N_a$ and environment keys $N_e$, and (c) the use of Farthest Point Sampling (FPS) and Multi-Level Contrast (MLC).

## 5 LIMITATIONS

While our simulation experiments demonstrate that ICon improves the base policy across a variety of manipulation tasks, our work has several limitations. First, our method is compatible only with vision transformers and their variants, which restricts its applicability to other commonly used visual encoder architectures in visuomotor policy learning, such as ResNet (He et al., 2016). Second, the Farthest Point Sampling (FPS) process incurs substantial computational overhead during forward propagation, making ICon inefficient for policy training. Eventually, our experiments are confined to simulation, and we have not yet evaluated our method in real-world settings due to limited hardware resources.

## 6 DISCUSSION AND FUTURE WORK

In this work, we investigate the benefits of grounding bodily awareness in visual representations and introduce ICon, a contrastive learning framework for extracting agent-centric representations from pixel observations. We demonstrate that policies augmented with ICon consistently achieve performance improvements across a diversity of manipulation tasks and can be effectively transferred across robots with different morphologies and configurations. In our future work, we plan to evaluate our method in complex real-world settings, where additional noise and distractors are present in the environments. Additionally, we hope to further enhance the learned agent-centric representations and develop more effective ones to enable cross-embodiment policy transfer.

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

## A  PSEUDOCODE FOR ICON

---

**Algorithm 2** Inter-token Contrast (ICon)

---

1: **Input:** an RGB image $\mathcal{I} \in \mathbb{R}^{H \times W \times 3}$, an agent mask $\mathcal{M} \in \mathbb{R}^{H \times W}$, a vision transformer $\mathcal{E}(\cdot)$ with patch size $P$ and embedding dimension $D$, number of agent-specific keys $N_a$, number of environment-specific keys $N_e$

2: **Output:** a contrastive loss $\mathcal{L}_{\text{ICon}}$

3: $[\mathcal{F}_{\text{cls}}, \mathcal{F}] \leftarrow \mathcal{E}(\mathcal{I})$

4: $\mathcal{F}_{\text{map}} = \{f_{k,l} \in \mathbb{R}^D\}_{k=1,l=1}^{H/P, W/P} \leftarrow \text{Reshape}(\mathcal{F})$

5: $\mathcal{P}_{\text{mask}} = \{p_{k,l}\}_{k=1,l=1}^{H/P, W/P} \leftarrow \text{Patchify}(\mathcal{M})$

6: $\mathcal{M}_{\text{token}} = \{m_{k,l} \in \{0,1\}\}_{k=1,l=1}^{H/P, W/P} \leftarrow \text{Threshold}(\mathcal{P}_{\text{mask}})$      ▷ Equation (2)

7: $q_a, q_e \leftarrow \text{Average}(\mathcal{F}_{\text{map}}, \mathcal{M}_{\text{token}}), \text{Average}(\mathcal{F}_{\text{map}}, 1 - \mathcal{M}_{\text{token}})$      ▷ Equation (3)

8: $\mathcal{K}_a, \mathcal{K}_e = \leftarrow \text{FPS}(\mathcal{F}_{\text{map}}, \mathcal{M}_{\text{token}}, N_a), \text{FPS}(\mathcal{F}_{\text{map}}, 1 - \mathcal{M}_{\text{token}}, N_e)$      ▷ Algorithm 1

9: $\mathcal{L}_a, \mathcal{L}_e \leftarrow \mathcal{L}_{\text{InfoNCE}}(q_a, \mathcal{K}_a, \mathcal{K}_e), \mathcal{L}_{\text{InfoNCE}}(q_e, \mathcal{K}_e, \mathcal{K}_a)$      ▷ Equation (1)

10: $\mathcal{L}_{\text{ICon}} \leftarrow \mathcal{L}_a + \mathcal{L}_e$

11: **return** $\mathcal{L}_{\text{ICon}}$

---

## B  IMPLEMENTATION DETAILS

### B.1  DATA AUGMENTATION

Following Chi et al. (2023), we apply random cropping to both RGB images and agent masks during training. The crop size is fixed at $3 \times 224 \times 224$ across all tasks. During inference, a static center crop of the same size is used.

### B.2  MODEL ARCHITECTURE

The policy networks used in this work are built upon the Diffusion Policy (Chi et al., 2023). We keep the overall model architecture unchanged except for the visual encoder, where we replace the ResNet (He et al., 2016) with a Vision Transformer (ViT) (Dosovitskiy et al., 2020). To save computing resources, we employ ViT-S with a patch size of 16 and an input image size of 224 as the visual encoder for our policy network.

### B.3  ENVIRONMENT SETUP

Details of the environment setup for RLBench and Robosuite are provided in Table 4. Note that in RLBench, robot proprioception includes arm joint positions, end-effector poses, and gripper status, whereas in Robosuite, robot proprioception consists of end-effector poses and gripper joint positions.

Table 4: Summary of task environments. **Objs**: number of objects in the scene; **Views**: number of viewpoints; **Img-Size**: image size; **P-D**: robot proprioception dimension; **A-D**: action dimension; **Controller**: robotic arm controller; **Steps**: maximum number of rollout steps.

| | Objs | Views | Img-Size | P-D | A-D | Controller | Steps |
|---|---|---|---|---|---|---|---|
| Close Microwave | 1 | 2 | $3 \times 256 \times 256$ | 14 | 7 | IK Pose | 150 |
| Close Drawer | 1 | 2 | $3 \times 256 \times 256$ | 14 | 7 | IK Pose | 200 |
| Open Box | 1 | 2 | $3 \times 256 \times 256$ | 14 | 7 | IK Pose | 200 |
| Take Lid off Saucepan | 2 | 2 | $3 \times 256 \times 256$ | 14 | 7 | IK Pose | 200 |
| Put Rubbish in Bin | 4 | 2 | $3 \times 256 \times 256$ | 14 | 7 | IK Pose | 300 |
| Lift | 1 | 2 | $3 \times 256 \times 256$ | 9 | 7 | OSC Pose | 200 |
| Door | 1 | 2 | $3 \times 256 \times 256$ | 9 | 7 | OSC Pose | 300 |
| Stack | 2 | 2 | $3 \times 256 \times 256$ | 9 | 7 | OSC Pose | 300 |

## B.4 TRAINING

We train our policy networks, ICon-Diff-C and ICon-Diff-T, using 3 training seeds (0, 42, and 100) and a batch size of 64. For each task, all policies are trained for 600 epochs on a single Nvidia GeForce RTX 3090 GPU, while in cross-robot transfer settings, the pre-trained policies are fine-tuned on the target robotic data for an additional 300 epochs. All other training configurations follow the settings described in the original codebase of Diffusion Policy (Chi et al., 2023).

## C VISUALIZATION OF LEARNED REPRESENTATIONS

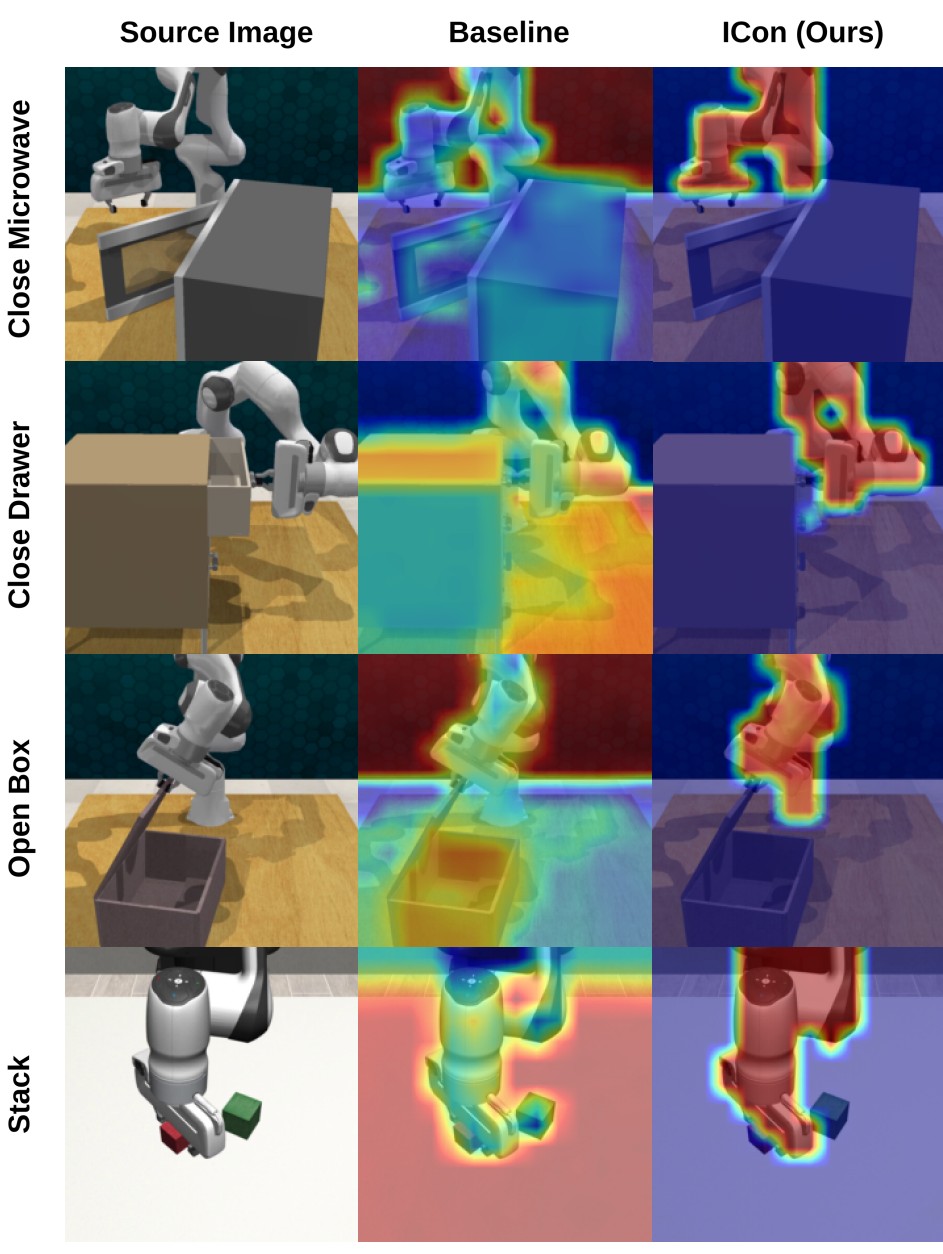

Figure 6: Visualization of representations learned by different algorithms across several tasks. For each task, we show the original image alongside the feature maps produced by different algorithms. Each feature map is computed by averaging the attention maps from all heads in the final layer of the vision transformer, with the [CLS] token as the query.

After training the vision transformer end-to-end with the policy network from scratch, we visualize the attention maps from the final layer of the vision transformer across several tasks. As shown in Figure 6, unlike the dispersed attention patterns exhibited by the baseline method, our contrastive learning approach encourages the vision transformer to focus on the agent's body rather than the entire scene. This confirms that the learned representations are agent-centric and carry body-relevant information about the robotic agent.

