# OpenReview forum: "Grounding Bodily Awareness in Visual Representations for Efficient Policy Learning"
_ICLR.cc/2026/Conference — Submitted to ICLR 2026_

### Official Review · Reviewer_CEhB · 2025-10-21

**Soundness:** 2
**Presentation:** 3
**Contribution:** 2
**Rating:** 2
**Confidence:** 4

**Summary:**

This work proposes an auxiliary objective for end-to-end learning of visual policies which employ a vision-transformer image encoder. The authors take a contrastive learning approach for encouraging agent-environment disentanglement in the learned ViT features. The objective leverages a segmentation mask of the agent (acquired with SAM) to classify ViT patches to either agent or environment, and employs an inter-token contrastive loss based on these classes. The method is evaluated on imitation learning by training visual diffusion policies on simulated robotic manipulation environments.

**Strengths:**

**Overview**
- Well written paper.
- Method seems novel.
- Potentially applicable to any policy that employs a ViT image encoder.
- Evaluated on numerous environments.

I am willing to raise my score if the points raised in the Weaknesses and Questions sections are addressed.

**Weaknesses:**

**Overview**
- The specific method is not well-motivated other than the agent-environment disentanglement.
- No comparison with other segmentation-based representation learning objectives.
- Information loss in produced token masks compared to the original pixel mask.
- Empirical performance gains are not significant.

**Method Motivation**

Why is your specific approach better than others for acquiring agent-environment disentanglement in feature space? This is neither discussed nor empirically evaluated in this work. You cited many papers that apply the same agent-centric principles, could you provide comparisons with one or more of these methods?

Additionally, I think it would be beneficial to perform the following ablations to motivate your contrastive approach:
1. Replacing the agent-mask-based contrastive objective with an agent-mask-based reconstruction objective. (Correct me if I am mistaken, but you are currently only comparing with a full-scene reconstruction baseline).
2. Replacing the contrastive objective with a simple inductive bias: applying a learned additive embedding to the patch features based on their association to environment vs. agent (total of 2 learned embeddings). This suggestion is based on the same principles as your contrastive approach---distinguishing between agent and environment features. If this is indeed beneficial for downstream policy performance, maybe it will be learned end-to-end, i.e., the embeddings will converge to be dissimilar?

**Token Masks**

The token mask threshold results in a coarser segmentation than the original pixel-based segmentation. Why is this preferred over, e.g., processing the masked image with two parallel ViT layers, once with the agent masked out and once with the environment masked out? The rest of the pipeline can stay the same after this initial layer by combining the tokens from both images such that they also attend to each other in the following layers.

**Experiments**

4/5 RLBench, 2/3 Robosuite and 1/2 transfer tasks are within a standard deviation from the base diffusion policy. It is standard to highlight in bold results that are within a standard deviation from the best performing method for clarity. Maybe evaluating with a larger number of seeds will help distinguish the performance gain. Currently, the performance gains look marginal at best and do not justify the additional method complexity.

**Questions:**

- Hyperparameters should be detailed in the Appendix. Specifically, what are the values of gamma and lambda? Do they have to be tuned per-task or is the algorithm robust to these hyperparameters?
- How robust is the method with respect to hyperparameters in terms of the training stability evaluated in Section 4.5?
- Figure 6: Why does the CLS token specifically attend only to the agent? What do the other tokens attend to?

---

> ### Author Response · Authors · 2025-11-24
> **Rebuttal by Authors**
>
> Thank you so much for the insightful and thoughtful feedback. We will address your questions below.
>
> > No comparison with other segmentation-based representation learning objectives.
>
> > Why is your specific approach better than others for acquiring agent-environment disentanglement in feature space? This is neither discussed nor empirically evaluated in this work. You cited many papers that apply the same agent-centric principles, could you provide comparisons with one or more of these methods?
>
> Compared with prior works [1, 2], a key advantage of our approach is that it **does not require additional decoder networks** for reconstructing images or masks. This substantially reduces computational overhead during model training. Unfortunately, we are unable to provide empirical comparisons with these methods because their model architectures (CNN encoders with MLP policies) differ significantly from ours (ViT encoder with a diffusion-based policy). We attempted to adapt the method in [1] to our setting by replacing the original CNN and MLP components with ViTs and diffusion policies. However, the resulting policies performed far below the baseline diffusion policy, which we believe is due to the incompatibility of the original design with our pipeline.
>
> [1] Efficient RL via Disentangled Environment and Agent Representations, ICML 2023.
>
> [2] DEAR: Disentangled Environment and Agent Representations for Reinforcement Learning without Reconstruction, IROS 2024.
>
> > Information loss in produced token masks compared to the original pixel mask.
>
> Thank you for bringing this up. We agree that converting pixel-level masks into token-level masks inevitably introduces some information loss due to the approximation involved. However, prior work [1] has shown that natural images exhibit substantial spatial redundancy, suggesting that fine-grained pixel details are often not required for extracting high-level semantic information. Based on this insight, we believe that aggregating pixel-level masks into token-level masks does not significantly impact the model’s ability to capture the essential agent–environment semantics.
>
> [1] Masked Autoencoders Are Scalable Vision Learners, CVPR 2022.
>
> > Additionally, I think it would be beneficial to perform the following ablations to motivate your contrastive approach: 1. Replacing the agent-mask-based contrastive objective with an agent-mask-based reconstruction objective. (Correct me if I am mistaken, but you are currently only comparing with a full-scene reconstruction baseline). 2. Replacing the contrastive objective with a simple inductive bias: applying a learned additive embedding to the patch features based on their association to environment vs. agent (total of 2 learned embeddings). This suggestion is based on the same principles as your contrastive approach---distinguishing between agent and environment features. If this is indeed beneficial for downstream policy performance, maybe it will be learned end-to-end, i.e., the embeddings will converge to be dissimilar?
>
> Thank you for the valuable comment. Regarding ablation 1, we implemented a variant of our pipeline that reconstructs the agent mask from ViT features using an image decoder similar to that in [1]. Interestingly, this modification improved the success rate on the Lift task (single seed), as shown below:
>
> | Task / Model | ICon | Mask Recons |
> |--------------|------|-------------|
> | Lift         | 0.72 | **0.76**    |
>
> We will evaluate this approach on additional tasks and training seeds to further assess its effectiveness.
>
> Regarding ablation 2, we are unsure how to determine or measure the association between the learned embeddings and the agent/environment. Could you please clarify this?
>
> [1] Crossway Diffusion: Improving Diffusion-based Visuomotor Policy via Self-supervised Learning. ICRA 2024
>
> > The token mask threshold results in a coarser segmentation than the original pixel-based segmentation. Why is this preferred over, e.g., processing the masked image with two parallel ViT layers, once with the agent masked out and once with the environment masked out? The rest of the pipeline can stay the same after this initial layer by combining the tokens from both images such that they also attend to each other in the following layers.
>
> Thank you for the question. Processing two masked images (agent-masked and environment-masked) requires running both through the ViT, which doubles the token length and significantly increases computational cost. Moreover, this design would still require agent masks at inference time, which is inefficient and impractical during policy rollout. In contrast, our method avoids any mask usage at inference and keeps the token budget fixed, making it far more efficient and deployable in practice.

---

> ### Author Response · Authors · 2025-11-24
> **Rebuttal by Authors (Continued)**
>
> > Hyperparameters should be detailed in the Appendix. Specifically, what are the values of gamma and lambda? Do they have to be tuned per-task or is the algorithm robust to these hyperparameters?
>
> The value of $\gamma$ is set to 0.1 and $\lambda$ to 1.0 for all tasks, except for the Lift task where we use $\gamma = 0.01$. Overall, we found the method to be robust to these hyperparameters, and no per-task tuning was required beyond this minor adjustment.
>
> > How robust is the method with respect to hyperparameters in terms of the training stability evaluated in Section 4.5?
>
> Thank you for the question. Unfortunately, we did not conduct a systematic evaluation of how different hyperparameter settings affect training stability. In our experiments, we found that the default hyperparameters worked reliably across most tasks, but a more thorough stability analysis would certainly be valuable for future work.
>
> > Figure 6: Why does the CLS token specifically attend only to the agent? What do the other tokens attend to?
>
> We believe this is because our method encourages the ViT to learn agent-centric representations, which in turn causes the attention mechanism to put more focus on agent-specific tokens.

---

> > ### Comment · Reviewer_CEhB · 2025-11-24
> >
> > Thank you for your response and effort in running additional experiments to address my questions and concerns.
> >
> > **Baselines/Ablations**
> >
> > I understand that it may be difficult to apply some previous related work directly as baselines due to various reasons. I still think that a segmentation-based baseline is missing. An ablation of your method such as the one proposed in my original review (ablation 1) which you have implemented could serve as that baseline.
> >
> > Regarding ablation 2, what I am proposing is to use the same masks used for your contrastive method in order to determine which learned embedding to add to each patch  based on the affiliation to agent or environment. The learned embeddings would simply be 2 different learned parameter vectors with a dimension equal to the patch feature dimension, 1 for agent patches and one for environment patches. I understand that this would require using the masks at test-time but this could still be a valuable ablation to demonstrate the strengths of your approach compared to a simpler method.
> >
> > Regarding the need for decoder networks, I agree that decoders can add overhead during training. It seems that some aspects of your contrastive method, such as the FPS, also add computational overhead. It would be valuable to quantify these and compare them in order to solidify this point.
> >
> > **Token Masks**
> >
> > My concern regarding the information loss due to token masks is not related to the spatial redundancy but to the fact that a given token may have critical information on *both the agent and the environment*. An example of this is a robotic gripper holding an object, in that case both the gripper and the object are key features for control but the two will not be distinguished by your method.
> >
> > **Hyperparameter Sensitivity**
> >
> > Thank you for providing further information about hyperparameters. I believe a hyperparameter sensitivity analysis would be beneficial to back-up the robustness claims.
> >
> > **Summary**
> >
> > Your responses have addressed some of my concerns/questions but several key concerns still remain:
> > - Method motivation is not sufficiently backed-up by experiments.
> > - Segmentation-based baseline results on all environments/tasks are missing.
> > - Empirical performance gains are not significant.
> >
> > I appreciate the time you have taken to respond to my review. Unfortunately, due to the above points, I cannot recommend acceptance of the paper in its current form.

---

### Official Review · Reviewer_GJDi · 2025-10-21

**Soundness:** 3
**Presentation:** 2
**Contribution:** 2
**Rating:** 6
**Confidence:** 4

**Summary:**

The paper presents a (class-supervised) contrastive representation learning approach of tokens from a Vision Transformer, termed Inter-Token Contrast (ICon), for the purpose of robotic manipulation. The core idea is to segment-out tokens relevant to the agent (e.g., the robotic gripper) and contrast them with tokens representing everything else. The method is implemented within a diffusion-based policy, and demonstrated on various tasks and robots from two simulated benchmarks.

**Strengths:**

* The utilization of FPS over 2D feature/token maps to ensure coverage is interesting.
* I appreciate the stability analysis.
* Nice video visualizations on the project website.
* Open-source data and code!

**Weaknesses:**

* The method relies on a pre-trained supervised segmentation model. While it is true that these models are performing very well, they are still prone to wrong detection or missing objects for unseen data.
* The masking procedure relies on heuristic (class thresholding) where it is unclear what would happen if part of the gripper and a small object (e.g., a small block) occupy the same tokens.
* I found the multi-level contrastive loss description in L246-251 confusing: I don’t understand how $\gamma$ is tuned per-layer and I also don’t understand the reasoning for setting $\gamma >0$ for deeper and shallower layers. Given the reason provided in the paragraph, I would expect something like “for shallower layers $\gamma$ is larger/smaller than deeper layers as the earlier layers produce more entangled representations”.
* The computational cost is unclear. For training, masks are pre-computed (I assume this is because the segmentation model, SAM, is large), FPS can also be time-consuming, and memory-wise, the ViT encoder needs to be probed at several layers, saving the features for the contrastive loss. What about inference time? Are observations masked on the fly?
* Overall, except for 3 tasks, it seems that the improvement is marginal compared to the baselines, and for some tasks all methods fall short. While I appreciate the authors chose hard benchmarks, given my assumed computational cost, it is hard to get convinced regarding the contribution (and given that masking has already been proven to be useful in decision making, also referenced in the related work section).

**Questions:**

* ViT: it is unclear from the text if the ViT encoder is pre-trained, fine-tuned or trained from scratch. Can you please clarify this?
* Regarding the masking procedure concern I raised under Weaknesses, can the authors please clarify what would happen in multi-object environments where objects and gripper share tokens?
* Can the authors please clarify my concern regarding the computational cost under Weaknesses?
* Are policies multi-task or trained per-task? Are they goal-conditioned?
* The authors mention they use action chunking (“Temporal Ensemble”), to my understanding, and that it was critical for the performance. Can you clarify this and what is the chunk size used in practice?
* I’m not sure I understand this sentence in the conclusion (L469-470) “restricts its applicability to other commonly used visual encoder architectures in visuomotor policy learning, such as ResNet”. Why is that? Basically, it is possible to extract 2D features maps (that represent the regions in the corresponding input image) and apply downsampled masks over them. What prohibits you from applying the contrastive loss on these features? Or do you mean the downstream processing of the post-contrasted features?
* Have the authors experimented with other types of self-supervised loss, perhaps ones that do not require negative samples (e.g., BYOL, [Grill, Jean-Bastien, et al. "Bootstrap your own latent-a new approach to self-supervised learning." Advances in neural information processing systems 33 (2020): 21271-21284.](https://arxiv.org/abs/2006.07733))

Overall, I raised several concerns regarding the method as detailed above. I’m willing to increase my score given convincing answers and clarifications to my review.

---

> ### Author Response · Authors · 2025-11-19
> **Rebuttal by Authors**
>
> Thank you so much for the insightful and thoughtful feedback. We will address your questions below.
>
> > The method relies on a pre-trained supervised segmentation model. While it is true that these models are performing very well, they are still prone to wrong detection or missing objects for unseen data.
>
> Thank you for the thoughtful comment. To mitigate potential errors, we manually verify the quality of all generated masks and remove any samples with incorrect detections or missing agent regions before training. This ensures that segmentation noise does not affect the learning process.
>
> > The masking procedure relies on heuristic (class thresholding) where it is unclear what would happen if part of the gripper and a small object (e.g., a small block) occupy the same tokens.
>
> > Regarding the masking procedure concern I raised under Weaknesses, can the authors please clarify what would happen in multi-object environments where objects and gripper share tokens?
>
> Thank you for bringing this up and we acknowledge that this scenario represents a potential failure case of our method. When part of the gripper and a small object occupy the same token, the class-thresholding heuristic may misclassify the token as either agent-specific or environment-specific, which could hinder contrastive learning. A potential mitigation strategy is to reduce the patch size so that overlap between the gripper and nearby objects is minimized. However, this comes at the cost of increased computational overhead, so an appropriate trade-off must be considered.
>
> > I found the multi-level contrastive loss description in L246-251 confusing: I don’t understand how is tuned per-layer and I also don’t understand the reasoning for setting $\gamma > 0$ for deeper and shallower layers. Given the reason provided in the paragraph, I would expect something like “for shallower layers $\gamma$ is larger/smaller than deeper layers as the earlier layers produce more entangled representations”.
>
> Thank you for pointing out this confusion. As defined in Equation (5), the multi-level contrastive loss is computed as a weighted sum of the inter-token contrastive losses across different ViT layers. The contribution of layer $i$ is scaled by the factor $\frac{\exp{(\gamma\cdot i)}}{\sum_i\exp{(\gamma\cdot i)}}$. When $\gamma > 0$, this term increases with $i$, which means that deeper layers receive larger weights while shallower layers receive smaller weights. This is the mechanism by which the loss is tuned per layer.
>
> > The computational cost is unclear. For training, masks are pre-computed (I assume this is because the segmentation model, SAM, is large), FPS can also be time-consuming, and memory-wise, the ViT encoder needs to be probed at several layers, saving the features for the contrastive loss. What about inference time? Are observations masked on the fly?
>
> > Can the authors please clarify my concern regarding the computational cost under Weaknesses?
>
> We report computational cost in terms of GPU hours required for training. All models are trained for 600 epochs on a single NVIDIA GeForce RTX 3090 GPU. The results are shown below:
>
> | Task / Model          | ICon-Diff-C | ICon-Diff-T |
> |-----------------------|-------------|-------------|
> | Close Microwave       | 9.46        | 9.57        |
> | Close Drawer          | 11.50       | 13.02       |
> | Open Box              | 17.28       | 17.47       |
> | Take Lid off Saucepan | 8.69        | 12.13       |
> | Put Rubbish in Bin    | 15.06       | 15.88       |
> | Lift                  | 9.11        | -           |
> | Door                  | 16.48       | -           |
> | Stack                 | 11.76       | -           |
>
> Regarding inference time, note that ICon operates **only during training** and does not introduce any additional computational overhead at inference. During rollout, the policy consumes raw observations without requiring agent masks. The inference latency is approximately **200 ms on CPU**, consistent across our model and all diffusion policy baselines.
>
> > ViT: it is unclear from the text if the ViT encoder is pre-trained, fine-tuned or trained from scratch. Can you please clarify this?
>
> The ViT encoder is **trained from scratch** on in-domain robotic data for both our method and all baselines. This setup partly explains the lower performance on certain tasks, as ViTs are known to be difficult to optimize without large-scale pretraining.
>
> > Are policies multi-task or trained per-task? Are they goal-conditioned?
>
> All policies are trained **per task** and are **not goal-conditioned**.

---

> ### Author Response · Authors · 2025-11-19
> **Rebuttal by Authors (Continued)**
>
> > The authors mention they use action chunking (“Temporal Ensemble”), to my understanding, and that it was critical for the performance. Can you clarify this and what is the chunk size used in practice?
>
> Thank you for the question. We provide a brief clarification of action chunking and temporal ensemble below. For additional details, we refer the reviewer to [1].
>
> - **Action Chunking:** At each timestep, instead of predicting only the next action, the policy predicts the next $k$ future actions (where $k$ is the chunk size). This helps mitigate compounding errors commonly seen in imitation learning.
> - **Temporal Ensemble:** During inference, the policy is queried at every timestep to produce chunked action predictions. Predictions corresponding to the same execution timestep are then fused via a weighted average, and the resulting action is executed by the robot. This improves action smoothness and helps prevent jerky motions.
>
> In practice, following the setup in Diffusion Policy [2], we use a chunk size of **16** for the CNN-based diffusion policy and **10** for the transformer-based diffusion policy. We also adopt the receding horizon control used in Diffusion Policy, so only a subset of the predicted action sequence is actually executed.
>
> [1] Learning Fine-Grained Bimanual Manipulation with Low-Cost Hardware, RSS 2023.
>
> [2] Diffusion Policy: Visuomotor Policy Learning via Action Diffusion, RSS 2023
>
> > I’m not sure I understand this sentence in the conclusion (L469-470) “restricts its applicability to other commonly used visual encoder architectures in visuomotor policy learning, such as ResNet”. Why is that? Basically, it is possible to extract 2D features maps (that represent the regions in the corresponding input image) and apply downsampled masks over them. What prohibits you from applying the contrastive loss on these features? Or do you mean the downstream processing of the post-contrasted features?
>
> Thank you for the constructive comment. Indeed, we have experimented with applying our method to CNN-based encoders. As you pointed out, the pixel-level masks are downsampled either through fixed interpolation or a learned projection, and then applied to the 2D CNN feature maps. However, we found that the resulting performance was below expectation. Our intuition is that the contrastive objective benefits from the tokenized and globally receptive representations produced by ViTs, where each token corresponds to a semantically meaningful region of the image. In contrast, CNN feature maps are spatially entangled and heavily influenced by local receptive fields. When masks are downsampled and applied to CNN features, the resulting masked regions become less semantically coherent, which weakens the contrastive supervision signal. Consequently, the improvements brought by our method do not consistently transfer to CNN architectures.
>
> > Have the authors experimented with other types of self-supervised loss, perhaps ones that do not require negative samples (e.g., BYOL, Grill, Jean-Bastien, et al. "Bootstrap your own latent-a new approach to self-supervised learning." Advances in neural information processing systems 33 (2020): 21271-21284.)
>
> Unfortunately, we did not experiment with self-supervised losses that do not require negative samples. However, this is indeed a promising direction for future work, as methods such as BYOL have been shown to produce stable and robust representations without the challenges associated with hard negative sampling. Integrating such objectives could potentially improve training stability and help address the issue of gripper–object overlap within the same token raised by the reviewer.

---

> > ### Comment · Reviewer_GJDi · 2025-11-22
> > **Thank you for the clarifications**
> >
> > I thank the authors for their clarifications and their effort during the rebuttal period. The authors have clarified my questions and I also acknowledge that I have read the other reviews. Overall, the idea of using FPS is interesting as I indiciated in my original review. However, the main concerns regarding this method remain: reliance on mask supervision, underperformance of the vision part due to training ViT from scrath, manual labor in engineering (such as manual removal of detections) and heuristics, and the marginal performance improvements.
> >
> > I will keep my score and I will not argue against rejection should the other reviewers vote rejection. Thanks again for your effort and clarifications.

---

### Official Review · Reviewer_kaVz · 2025-10-31

**Soundness:** 2
**Presentation:** 3
**Contribution:** 2
**Rating:** 2
**Confidence:** 3

**Summary:**

The paper introduces ICon, a simple contrastive objective that encourages ViTs to disentangle representations of the agent’s body from the environment in visuomotor RL. Using pre-extracted masks of the agent's body, the authors use farthest-point sampling to sample tokens which correspond to the agent's body and to the environment and appliy an InfoNCE loss, as an auxiliary loss for Diffusion Policy. The paper is clearly written and the approach to solve the problem is technically sound, although the usage of hard positive/negatives of agent/environment in the contrastive learning part might introduce some problems.
The experiments of the authors on RLBench and Robosuite do show that ICon leads to improvements on downstream tasks and some cross-robot transfer. However, the experimental section could be strengthened to better showcase the strenght of the learned representations.

**Strengths:**

1) The approach to disentangle agent-specific and environment specific features is technically sound. The FPS also helps obtain more useful representations w.r.t. random sampling.
2) The improvement in success rates for a few tasks in Robosuite and RLBench seems to be consistent.
3) With little finetuning the method seems to easily transfer across different robots although it is not clear how much tuning is needed and how well the method would perform zero-shot.

**Weaknesses:**

* The benchmarks used to validate the method are limited. Training concurrently ViT with the policy also seems to limit the ability to do some pretraining. I believe it would be useful if the objective was also used not only concurrently with the policy, but with a large pretraining setup that can later be used out of the box for variosu downsteram tasks without tuning.

* How much fine-tuning is needed ? Also would it work out of the box ? I believe that at its current state since we train on one robot at each time the results do not generalize directly.

* The method as the authors state is tied to ViTs and needs agent masks. This is however not a major weakness since ViT are currently the standard go-to for visual representations.

* The gains are consistent but also are small. It would be nice also if the authors included more baselines with popular pretrained encoders (such as CLIP).

* The training stability and ablation studies are all done in different environment at each time. This is weird - please use a single unified framework for doing your ablation studies and stability evaluation.

* The overhead of using FPS is not quantified.

**Questions:**

1) I believe that the current direction of the authors is useful and intuitive and it would make sense to help robotic agents disentangle features from their body and the environment. In my opining the object the authors propose would be also very useful in pretraining of ViTs to use in downstream tasks afterwards. This can allow to train on multiple robotic agents and generalize out of the box to new ones.

2) What is the overhead of using FPS ? How would it compare with dense sampling of tokens ? Also, how would it compare with some simple uniform grid sampling ?

3) Please use a common unified ablation framework.

4) How sensitive is your method in the mask extracted for the agent. Have you done any studies on this ?

5) One major question and a problem with contrastive learning is the usage of hard negatives. At its current state the positive=agent, negative=environment pair is a bit blunt and can worsen some representations since some features from the environment might be similar with the agents (e.g. texture or color). Have the authors consider an alternative to contrastive learning ? Would it be better maybe to force some features and not all features in the feature vector of the token to be distant and not all at once ?

---

> ### Author Response · Authors · 2025-11-19
> **Rebuttal by Authors**
>
> Thank you so much for the insightful and thoughtful feedback. We will address your questions below.
>
> > The benchmarks used to validate the method are limited. Training concurrently ViT with the policy also seems to limit the ability to do some pretraining. I believe it would be useful if the objective was also used not only concurrently with the policy, but with a large pretraining setup that can later be used out of the box for variosu downsteram tasks without tuning.
>
> > The method as the authors state is tied to ViTs and needs agent masks. This is however not a major weakness since ViT are currently the standard go-to for visual representations.
>
> > I believe that the current direction of the authors is useful and intuitive and it would make sense to help robotic agents disentangle features from their body and the environment. In my opining the object the authors propose would be also very useful in pretraining of ViTs to use in downstream tasks afterwards. This can allow to train on multiple robotic agents and generalize out of the box to new ones.
>
> Thank you for raising these points. We agree that leveraging the objective for large-scale pretraining would be highly beneficial. However, current large-scale robotic datasets with comprehensive and reliable mask annotations across diverse tasks and environments remain limited, which prevents our method from scaling to a full pretraining regime at this stage. That said, recent progress in automated segmentation techniques is rapidly improving the feasibility of generating high-quality mask annotations from existing robotic data. We believe these advances will enable more scalable pretraining and further unlock the potential of our approach.
>
> > How much fine-tuning is needed ? Also would it work out of the box ? I believe that at its current state since we train on one robot at each time the results do not generalize directly.
>
> In the cross-robot transfer setting, fine-tuning both the image encoders and the policy network requires **5** demonstrations on the target robot and approximately **300 epochs** to achieve the results reported in Table 3. When only the policy network is fine-tuned (also with 5 demonstrations for 300 epochs), the corresponding success rates on the Lift task are as follows:
>
> | Model / Robot  | Franka (Source) | Kinova (Target) |
> |-------------|--------------------------|--------------------|
> | Diff-C      | 0.527 ± 0.104            | 0.167 ± 0.061      |
> | ICon-Diff-C | 0.627 ± 0.129            | 0.227 ± 0.031      |
>
> As you correctly noted, since the policy is trained on a single robot and a single task at each time, it does not fully generalize to unseen robots or environments without additional fine-tuning.
>
> > The gains are consistent but also are small. It would be nice also if the authors included more baselines with popular pretrained encoders (such as CLIP).
>
> Thank you for the valuable feedback. In response, we added comparisons with two additional baselines, **MoCo-Diff-C** and **MAE-Diff-C**, whose image encoders are pretrained on the same in-domain robotic dataset using MoCo-v3 [1] and MAE [2], respectively. We did not include CLIP-pretrained encoders because our dataset is single-task and does not include language supervision. Due to time constraints, we report results on the Lift task using a single training seed. The success rates are shown below:
>
> | Task / Model | MoCo-Diff-C | MAE-Diff-C | ICon-Diff-C |
> |--------------|-------------|------------|-------------|
> | Lift         | 0.66        | 0.68       | **0.72**    |
>
> These results indicate that even when compared against pretrained encoder baselines, our method still achieves higher performance.
>
> [1] An Empirical Study of Training Self-Supervised Vision Transformers, ICCV 2021.
>
> [2] Masked Autoencoders Are Scalable Vision Learners, CVPR 2022.
>
> > The overhead of using FPS is not quantified.
>
> > What is the overhead of using FPS? How would it compare with dense sampling of tokens? Also, how would it compare with some simple uniform grid sampling?
>
> To quantify the overhead of FPS, we report the GPU hours required to train our model under different sampling strategies. We compare FPS with a simple sampling baseline that randomly selects a fixed number of tokens. All experiments were conducted on the Close Microwave task using 50 demonstrations, trained for 600 epochs on a single NVIDIA GeForce RTX 3090 GPU. The results are summarized below:
>
> | Method / Agent & Env Tokens | 5, 25 | 10, 50 | 20, 100 |
> |---------------------|-------|--------|---------|
> | Random Sampling     | 7.253 | 7.328  | 7.455   |
> | FPS                 | 8.705 | 9.464  | 13.239  |
>
> The results indicate that FPS introduces additional computational overhead compared with simple random sampling, and the cost increases with the number of selected tokens.

---

> > ### Author Response · Authors · 2025-11-19
> > **Rebuttal by Authors (Continued)**
> >
> > > The training stability and ablation studies are all done in different environment at each time. This is weird - please use a single unified framework for doing your ablation studies and stability evaluation.
> >
> > > Please use a common unified ablation framework.
> >
> > Thank you for the valuable feedback. We acknowledge this oversight and will revise the experimental setup to ensure that all ablation studies and stability evaluation are performed consistently within a single, standardized environment.
> >
> > > How sensitive is your method in the mask extracted for the agent. Have you done any studies on this?
> >
> > We evaluated the sensitivity of our method to inaccuracies in the agent masks by replacing the original masks with three types of corrupted versions: (i) masks with missing regions, (ii) masks with salt-and-pepper noise, and (iii) masks with jagged boundaries. We observed that inaccurate masks typically reduced model performance by approximately 10–20% across different tasks. Among the three variants, the masks with jagged boundaries produced the largest degradation, likely because the irregular boundaries make it more difficult to separate agent-specific and environment-specific tokens, thereby hindering effective agent–environment disentanglement in the learned representations.
> >
> > > One major question and a problem with contrastive learning is the usage of hard negatives. At its current state the positive=agent, negative=environment pair is a bit blunt and can worsen some representations since some features from the environment might be similar with the agents (e.g. texture or color). Have the authors consider an alternative to contrastive learning ? Would it be better maybe to force some features and not all features in the feature vector of the token to be distant and not all at once?
> >
> > Thank you for the insightful comment. Following your suggestion, we experimented with applying contrastive learning only to a subset of feature dimensions using a learnable feature mask. The corresponding success rates on the Lift task (single training seed) are shown below:
> >
> > | Task / Method | Partial Features | Full Features |
> > |---------------|------------------|---------------|
> > | Lift          | 0.46             | **0.72**      |
> >
> > Unfortunately, restricting the contrastive objective to a partial feature subset did not improve performance on this task. Nonetheless, we agree that the use of hard negatives is an important concern, and we plan to explore alternative approaches to address this issue in future work.

---

### Meta-Review · Area_Chair_Rc1B · 2026-01-06

**Summary:**

The paper proposes "Inter-Token Contrast" (ICon), a method using contrastive learning on ViT tokens to disentangle agent and environment representations for robotic manipulation. While the writing is clear, the consensus among reviewers is that the paper is not ready for publication. The primary issues are the marginal empirical improvements, the significant reliance on heavy supervision (segmentation masks with manual filtering), and conceptual limitations regarding "hard negatives" during object interaction.

**Reviewer Concerns:**

Addressed:

1）Clarified computational costs (overhead of Farthest Point Sampling).

2）Added comparisons to MoCo and MAE pre-trained baselines.

3）Provided implementation details regarding action chunking.

Outstanding:

1）Marginal Gains: All reviewers noted that performance improvements are insignificant compared to baselines.

2）Conceptual Flaw (Hard Negatives): The contrastive objective forces separation between agent and environment features, which is detrimental when the gripper interacts with an object (sharing tokens).

3）Ablation Results: As noted by Reviewer CEhB, the authors' own rebuttal showed that a simpler "mask reconstruction" ablation outperformed the proposed method.

**Reviewer Scores:**

Reviewer kaVz: 2 (Reject) - Maintained.

Reviewer GJDi: 6 (Marginally above) - Stated they will not argue against rejection; concerns about supervision and marginal gains remain.

Reviewer CEhB: 2 (Reject) - Maintained.

---

### Decision · Program_Chairs · 2026-01-26

Reject